# Safe and Effective Treatment of Patients with Urinary Tract Infections Caused by Extended-Spectrum Beta Lactamase-Producing Enterobacteriaceae via Telemedicine-Controlled Hospital at Home: A Case Series of 11 Patients

**DOI:** 10.3390/reports7020031

**Published:** 2024-04-26

**Authors:** Mayan Gilboa, Sholem Hack, Shahar Hochner, Mark Gitbinder, Megi Yakovlev, Noa Bineth, Galia Barkai, Gad Segal

**Affiliations:** 1Infection Prevention Unit, Sheba Medical Center, Ramat Gan 5262000, Israel; 2Faculty of Medicine, Tel Aviv University, Ramat Aviv, Tel Aviv 6997801, Israel; 3Education Authority, Chaim Sheba Medical Center, Ramat Gan 5262000, Israel; 4St. Georges University London School of Medicine, Program Delivered by University of Nicosia at the Chaim Sheba Medical Center, Ramat Gan 5262000, Israel; 5Tel Aviv Sourasky Medical Center, Tel Aviv 6423906, Israel; 6Faculty of Medicine and Dentistry, Palacky University Olomouc, 78371 Olomouc, Czech Republic; 7Department of Military Medicine, Faculty of Medicine, Hebrew University of Jerusalem, Jerusalem 9002002, Israel; 8Beyond Virtual Hospital, Chaim Sheba Medical Center, Ramat Gan 5262000, Israel

**Keywords:** urinary tract infection, ESBL, extended-spectrum beta lactamase, telemedicine, hospital at home, case series

## Abstract

Background: Resistant bacteria causing urinary tract infections (UTI) are becoming increasingly common worldwide. Patients suffering from such UTIs are often elderly, with complex medical backgrounds, and require prolonged hospital stays due to the frequent need for intravenous antibiotics. The alternative hospital-at-home (HAH) option for such patients should, therefore, be explored. Methods. We present our experience in the treatment of patients with extended-spectrum beta-lactamase (ESBL) infections treated through our HAH service. Results: Eleven such patients were included in our HAH service between February 2022 and December 2023 (median age: 79 years; 64% females; 57% had diabetes mellitus in their background). Of these patients, 27.2% had urinary instrumentations and 81.8% had a history of previous UTIs, of which 77.7% had resistant bacteria. The most common pathogen (7 out of 9 patients) was *Escherichia coli.* All eleven pathogens were resistant to ceftriaxone and ciprofloxacin. The mean length of hospitalization was 5 ± 2 days. Eight patients (72.7%) did not experience recurrent UTIs. Only two patients (18.2%) experienced acute kidney injury, which resolved during their HAH stay. Two patients died during a 30-day follow up from causes unrelated to their UTI. Conclusions: Treatment of patients presenting with urinary tract infections with resistant ESBL pathogens in the setting of a telemedicine-assisted, hospital-at-home setting is both effective and safe.

## 1. Introduction

### 1.1. In-Hospital Treatment of Urinary Tract Infections 

A urinary tract infection (UTI) is the invasion of the urinary system by pathogenic bacteria causing lower-urinary-tract-associated symptoms such as increased frequency, urgency, and dysuria or upper-urinary-tract-associated symptoms such as flank pain or fever. UTIs are one of the most common bacterial infections in humans, accounting for more than 150 million cases each year worldwide [1,2]. The yearly costs attributed to UTIs are estimated to be over USD 1.6 billion in the US alone. The main pathogens responsible for UTI are Gram-negative bacteria, most commonly *Escherichia coli*; other species such as *Klebsiella pneumoniae* and *Proteus mirabilis* (of the Enterobacteriaceae family); and, less commonly, Gram-positive pathogens, such as *Enterococcus faecalis* and *Staphylococcus saprophyticus* [3]. During the past several years, antimicrobial resistance has been steadily increasing, in large part due to the widespread use of wide-spectrum antibiotics [4]. The resultant emergence of extended-spectrum beta-lactamase (ESBL)-producing bacteria presents an increasing concern due to the resultant longer hospitalization durations and higher in-hospital complication rates [2,5,6]. The prevalence of resistance varies across different parts of the world, with some geographic locations experiencing rates of 50% and higher in ESBL-producing pathogens [2,3]. In these circumstances, some of the common regimens for the empiric treatment of UTI could prove ineffective and expensive [7]. Inappropriately treated lower-tract UTIs may progress into infections of the upper urinary tract, pyelonephritis, or urosepsis [8]. Complicated UTIs result in longer hospitalizations with more extensive use of antibiotic therapy, often associated with high mortality rates due to urosepsis [9]. The economic impact of ESBL-producing uro-pathogens is influenced by higher hospital costs due to longer hospitalizations and higher costs of antibiotic treatment [10]. Prolonged hospitalization also increases the rate of hospital-acquired complications, such as infections other than UTIs, venous thromboembolism (VTE), increased incidence of falls, and delirium [11].

### 1.2. Treatment of Infections in the Hospital-at-Home Setting

Home-based telemedicine-controlled hospitalization, the utilization of remote medical surveillance technologies for patients receiving advanced medical care in their domestic environment, is a recent medical advancement that became more prevalent during the COVID-19 pandemic [10], and is generally referred to as “Hospital-At-Home” (HAH). For many clinical scenarios, it is a viable substitute for inpatient hospitalization, and it is beneficial due to both reducing the cost of patient care and relieving hospital overcrowding [11,12,13]. Previous studies have shown that home hospitalizations of selected patients can be superior to conventional hospitalizations in that they lead to fewer readmissions, better quality of life, decreased incidence of delirium, as well as reduced overall medical complications [14,15]. Telemedicine-controlled home hospitalizations in the HAH setting enhance conventional home care by not just enabling the patient to stay at home, but also providing better remote monitoring managed by the patient or a family member. These technologies, such as the use of a simple six-lead electrocardiogram for assessing arrhythmias or remote auscultation of heart and lung sounds, have been previously described and validated [10,16]. The field of HAH is constantly evolving, and there is an ongoing need to assess various conditions and disease states that can be managed through this type of care. Some conditions that can be safely treated through HAH include chronic heart failure deterioration, COVID-19 infections, chronic obstructive pulmonary disease exacerbations, community-acquired pneumonia, soft tissue infections (both cellulitis and diabetic foot infections), and urinary tract infections [17]. 

### 1.3. Aim of the Current Study

In this study, we describe a cohort of patients who demonstrated positive urine culture results demonstrating multi-drug-resistant bacteria and were managed in HAH environments. The novelty of this research is in terms of a “proof of concept” that such patients, who are considered complicated and necessitating relatively high-maintenance treatment, could stay in their homes without compromising their safety or their chances of recovery. 

## 2. Methods

After the acceptance of ethical approval by our institutional review board (#SMC 0829-23), which waived the need for patients’ informed consent due to the retrospective nature of this study, we reviewed the electronic medical records of the Sheba-Beyond Virtual Hospital—the Telemedicine-based hospital-at-home arm of the Chaim Sheba Medical Center, the largest tertiary hospital in Israel. We found eleven patients that were hospitalized at their homes through this service due to ESBL-associated UTIs during a 22-month period.

Patients were referred to the HAH from the emergency department or internal medicine wards if they met the following criteria: All were hemodynamically stable, did not need further inpatient imaging, had a maximum of a twice-daily antibiotic requirement (in line with the service enabling up to two home visits a day), and had a supportive home environment with a caregiver present throughout the home hospitalization. All cases referred to HAH treatment were indicated by the attending physician for completion of intravenous antibiotic administration on the grounds of their pathogens’ antibiotics resistance profiles and intentions to prevent, as much as possible, the recurrence of infection. All eligible patients were assessed by an experienced physician specialized in internal medicine before approval for home hospitalization. Cases that were deemed appropriate to be treated by oral antibiotics were discharged from hospitalization and did not meet the eligibility criteria for HAH settings. 

Treatment during HAH included a once-daily telemedicine call with a specialized internal medicine physician and an in-person visit at least once daily from a trained nurse for vital signs measurements and administration of medications, as indicated. Additionally, there were two daily nurse-initiated telephone calls for follow-up and self-reporting of vital signs. Blood and urine tests, including creatinine, electrolytes, complete blood count with differential white blood cell count, cultures, and relevant drug levels, were obtained at the physician’s discretion. Antibiotics were administered during the nurse’s visit. A call center was also available for patients and family members to ask questions and report new symptoms.

A thorough retrospective manual review of medical records was conducted to exclude cases of false-culture results or irrelevant cases, such as patients hospitalized for less than one day. We describe these patients’ demographics, baseline morbidities, cause of acute hospitalizations, types of bacteria with antibiotic resistance, types of antibiotic treatment received, and clinical outcomes of both the hospitalization and the 30 days after discharge. 

For descriptive analysis, categorical variables (e.g., patients’ gender) are expressed as population percentages and continuous variables (e.g., patients’ age) as the mean ± standard deviation for normally distributed data, or otherwise as the median and interquartile range (IQR). 

## 3. Results

### 3.1. Patients’ Demographics and Pathogens

Between February 2022 and December 2023, we treated eleven patients with positive urine cultures for ESBL-producing Gram-negative bacteria through our HAH platform. Of these eleven, five had other, non-urinary, alternative primary diagnoses at admission (aspiration and bacterial pneumonia, pelvic osteomyelitis, and COVID-19 pneumonia). Table 1 describes our cohort patients’ characteristics. 

Patients had a median age of 79 years (IQR: 60 to 90 years), with a majority being female (7 out of 11 patients, 64%). Of the eleven patients presenting with positive urinary cultures for ESBL, only one patient had a positive blood culture result with the same pathogen as in his urine culture. Clinically, five had lower urinary tract infections, one was classified as suffering from an upper urinary tract infection, and the other five had alternative clinical diagnoses necessitating hospitalization. Relating to their medical background, only one had an impaired cognitive state, and the most common background chronic disease was diabetes mellitus (four out of eleven patients, 36.3%). Chronic urinary instrumentations (indwelling catheters or nephrostomies) were apparent in three patients (27.3%); nine (81.8%) had a history of past urinary tract infection; and most of them (7/9, 77.8%) already experienced a UTI with resistant bacteria in the past. 

Table 1 also presents the genera of bacterial growth in urinary cultures and the type of antibiotics to which the bacteria were either sensitive or resistant. The most common pathogen (seven out of nine patients, 77.8%) was *Escherichia coli.* All eleven pathogens were resistant to ceftriaxone and ciprofloxacin. All eleven were sensitive to piperacillin–tazobactam and meropenem. Seven (63.6%) were sensitive to gentamicin, while eight (72.7%) were found to be sensitive to amikacin.

Initially, all patients were treated at our medical center’s emergency department (ED). After early stabilization and initial diagnosis, most were admitted directly to our HAH service, while two (18.2%) were first admitted to an in-hospital internal medicine department and were transferred to our HAH service later. Upon presentation in the ED, six patients (54.5%) had urinary complaints, and three (27.3%) were febrile (temperature above 38 °C). During their HAH stays, two patients (18.2%) were diagnosed with acute kidney injury, defined as a blood creatinine level rising above 1.5 times the baseline value. All patients were considered stable in the ED and did not require administration of vasopressors or intensive fluid resuscitation. The antibiotic treatment was tailored to each patient’s specific characteristics and urine culture results, with variations among patients as detailed in Table 1. Patients who received aminoglycosides underwent daily drug-level monitoring for dosage adjustment as well as daily creatinine level measurement. Two patients were not treated with antibiotics at all since their clinical presentation, signs, and symptoms were not considered to be associated with their urinary culture results. Nevertheless, both of them were originally eligible for in-hospital stays, workup, and treatment, and therefore were not excluded from this series of patients even though they were not treated with antibiotics. 

### 3.2. Patients’ Clinical Outcomes

The mean hospitalization length was 5 ± 2 days. At discharge, all patients showed clinical improvement, including resolution of fever, and among those who were able to provide a history, improved symptoms of dysuria. Inflammatory markers such as white blood cell counts and differential and C-reactive protein blood concentrations improved in all patients. Of the eleven study patients, eight (72.7%) experienced no recurrent UTI within 30 days post-HAH discharge, while data relating to the three additional patients remain unknown. While formal surveys did not assess patient satisfaction, anecdotal evidence suggested high satisfaction levels, as patients and their families frequently expressed appreciation and positive feedback to the staff. To the best of our knowledge, two patients were re-admitted to our medical center for non-UTI causes, and two other patients died during a 30-day follow-up period due to non-UTI related-causes. 

## 4. Discussion

In this article, we describe the characteristics and outcomes of a cohort of patients receiving at-home hospitalization services while bearing highly resistant bacterial pathogens in their urinary tract samples. Some of them were indeed diagnosed as suffering from associated UTIs, and some were hospitalized due to other clinical entities. Outside the setting of this study, such patients were all considered eligible for in-hospital admission and were not intended to stay in their homes, in accordance with acceptable guidelines for HAH services worldwide. All patients were deemed appropriate, by their attending physicians at the hospital, to be treated with intravenous antibiotics. Patients that were deemed appropriate for oral antibiotics were discharged from hospitalization and did not meet the eligibility criteria for HAH settings. In part, the completion of intra-venous courses of antibiotics was considered in light of reports regarding the higher incidence of infection recurrence after oral therapy [18], while antibiotics that were shown to be effective in this setting, such as pivmecillinam, are not currently available in Israel currently. It is known, that ESBL-bearing patients are mostly elderly, with complex medical backgrounds, and accordingly, our patient group experienced a relatively high 30-day mortality rate (2 out of 11 patients, 18.2%), highlighting that, indeed, patients with resistant bacterial infections are at-risk individuals with a significant burden of chronic diseases [19,20,21,22]. The portion of patients with urinary tract instrumentations, either a simple urinary catheter or nephrostomy, was 3/11 (27.3%), which is also in concordance with what is known in the relevant literature [23]. However, considering the high-risk nature of this population, the outcomes of home hospitalization were generally favorable. The fact that only one patient had a positive blood culture also characterizes this patient population and the conclusions derived from our study, and, therefore, should not be applied to patients with ESBL bacteremia. 

The prevalence of ESBL-carrying patients requiring acute hospitalization is expected to rise [24]. Coupled with various factors contributing to a global crisis in the healthcare workforce and organizations, this underscores the necessity of establishing alternatives to in-hospital stays for such complex patients. In their 2016 systematic review, Conley et al. found that HAH settings are associated with mortality rates, disease-specific outcomes, and patient and caregiver satisfaction, which were either improved or not different compared with inpatient stays [25].

As detailed in the introduction to this manuscript, the HAH service in the current study provides a technologically enhanced, telemedicine-controlled, hospital-at-home environment. Sheba Beyond offers such services, but is continually obligated to demonstrate that it does not endanger acutely ill patients while offering them effective therapeutic solutions. In the current study, we aimed to summarize our experience with one such group of patients: patients suffering from UTIs with ESBL-producing bacteria. 

The results of our experience, although describing a cohort of patients without a comparison group, demonstrate the following lessons: (A). Treating resistant urinary infections in elderly, chronically ill patients in an HAH setting is safe. None of our patients died at home during the HAH days, and those who did die at the 30-day follow-up died of other, non-UTI-related causes. (B). Treatment of such patients at home with adequate antibiotics, while monitoring them clinically and with the appropriate laboratory tests, is effective and safe. All our patients recovered in relation to their clinical signs and symptoms (e.g., fever and dysuria), and all patients demonstrated improvements in their inflammatory markers (e.g., white blood cell counts, differential and C-reactive protein blood concentrations). Also, those patients who did experience acute kidney injury quickly recovered their kidney functions, and their tests returned to their baseline values prior to hospitalization. (C). To our knowledge, there were no cases of recurrent UTIs during the 30-day follow-up period. (D). Patient satisfaction, although not formally assessed, was very high, as is consistent with the general preference for home-based care over in-hospital stays.

The extensive availability of laboratory testing and vigilant monitoring by clinical teams enabled the treatment of patients with more complicated conditions, or those receiving treatments requiring closer follow-up, such as aminoglycosides, beyond what conventional home care could offer. It should be noted that we did not include a cost-effectiveness analysis in this study, and we intend to include financial analyses in future, larger studies in order to assimilate the service into available reimbursement programs. 

This article presents a small retrospective series of patients without comparison to a parallel control group. Therefore, there is room for a prospective, randomized clinical study comparing patients treated in an in-hospital setting versus HAH. Also, this was a single-center study; therefore, conclusions cannot be drawn relating to heterogeneous patient populations or different HAH services. We demonstrated the effectiveness and safety of HAH for patients with ESBL UTIs, but conclusions should not be drawn regarding patients with concurrent ESBL bacteremia. 

## 5. Conclusions

Treatment and follow-up of elderly, chronically ill patients bearing ESBL pathogens in their urine, either suffering from urinary tract infections caused by those resistant bacteria without bacteremia or suffering from a concurrent non-UTI clinical disease, are both safe and effective when delivered in the setting of a telemedicine-controlled hospital-at-home service. Such patients, when carefully selected, should be directed to HAH services which are capable of intravenous medication administration and close clinical and laboratory follow-up. Extension of such services and inclusion of ESBL-bearing patients would, potentially, partially alleviate the burden of patients over-crowding general, internal, and geriatric in-hospital departments. 

## Figures and Tables

**Table 1 reports-07-00031-t001:** Patients’ characteristics.

Patient #	1	2	3	4	5	6	7	8	9	10	11
Patients’ Demographics											
Gender (M/F)	F	F	F	F	M	F	F	F	M	M	M
Age (years)	89	25	79	76	94	90	90	60	74	88	45
Clinical Characteristics											
Diabetes (Yes/No/PRE)	Yes	No	Yes	Yes	No	No	No	No	Yes	No	No
U. instrument (Ne/C/PER)	No	No	C	No	No	No	No	No	C	No	Ne
Past UTI	Yes	Yes	Yes	Yes	Yes	No	No	Yes	Yes	Yes	Yes
Past res. UTI	Yes	No	No	Yes	Yes	No	No	Yes	Yes	Yes	Yes
Admission Dx. (LU/UU/O)	O	UU	O	LU	LU	O	O	O	LU	LU	LU
Bacterial Characteristics											
Positive blood cultures	No	No	No	No	No	No	No	Yes	No	No	No
Genus (E/P/K)	P	E	E	K	E	P	E	K	E	E	E
AKN (S/R)	S	S	S	S	I	S	S	I	S	I	S
GEN (S/R)	R	S	S	R	R	R	S	S	S	S	S
CEF (S/R)	R	R	R	R	R	R	R	R	R	R	R
PIP-TAZ (S/R)	S	S	S	S	S	S	S	S	S	S	S
MER (S/R)	S	S	S	S	S	S	S	S	S	S	S
CIP (S/R)	R	R	R	R	R	R	R	R	R	R	R
Hospitalization Characteristics											
HAH days	3	6	2	5	6	2	8	5	7	7	4
Abx home type	AKN	AKN	NONE	AKN	AKN	NONE	ERT	ERT	AKN	ERT	AKN
AKI (Yes/No)	Yes	No	No	No	No	No	Yes	No	No	No	No
Clinical Outcomes											
Recurrent UTI in 30 days (Yes/No)	No	No	No	No	No	No	NA	No	No	NA	NA
Re-admission, at 30 days	No	No	Yes	No	Yes	No	No	No	No	No	No
Death at 30 days	Yes	No	No	No	No	No	Yes	No	No	No	No

Gender: M = male, F = female; diabetes: PRE = pre-diabetes; urinary instrumentation: C = catheter, Ne = nephrostomy; admitting diagnosis: UU = upper UTI, LU = lower UTI, O = other; genus: E = *Escherichia coli*, P = *Proteus mirabilis*, K = *Klebsiella pneumoniae*; S = sensitive, R = resistant, I = intermediate; AKN = amikacin; GEN = gentamicin; CEF = ceftriaxone; PIP-TAZ = piperacillin–tazobactam; MER = meropenem; CIP = ciprofloxacin; ERT = ertapenem; Abx = antibiotic; AKI = acute kidney injury. NA = not available.

## Data Availability

The anonymized data of patients will be available upon reasonable request from the corresponding author.

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
