# Peer review of "Safe and Effective Treatment of Patients with Urinary Tract Infections Caused by Extended-Spectrum Beta Lactamase-Producing Enterobacteriaceae via Telemedicine-Controlled Hospital at Home: A Case Series of 11 Patients"

_reports, 2024, doi:10.3390/reports7020031_

Round 1
Reviewer 1 Report
Comments and Suggestions for Authors
Gilboa et al. present a rather small study concerning 11 patients treated at Hospital-at-Home for MDR UTI - two patients apparently were not considered to have UTI, but MDR bacteriuria was present. The concept is interesting in view of saving patients from in-hospital stay with high cost and risk of iatrogenic complications.
Comments:
1. First of all, the authors should discuss, why switch to oral antibiotics was not considered in any of the cases. Apparently, the patients were found healthy enough to leave the hospital and be followed at home, and assumingly - this is not quite clear from the text - the patients had been treated for UTI (except two) at hospital, again I guess, with iv antibiotics. In Northern Europe, where this reviewer resides, these patients would have been switched to oral drugs such as nitrofurantoin, fosfomycin or pivmecillinam, which are usually (at least one of them) active againt such pathogens.
2. It is not clear why the two patients, who were not considered to have UTI, were included in the study - it does not concur with the title of the study; unless good arguments, they should be deleted.
3. Considering the 9 patients remaining the format of the paper should be changed to letter or note.
4. One really misses line numbering to be able to refer to specific sites in the paper!
In the abstract line 6 from bottom the sentence. "Eight patients..." should be revised, does not make sense.
K. pneumoniae is spelled with an "e" at the end (several places, also in Table legend). And Gram-colour with capital G (name).
P. 4 lines 4 and 5: Antibiotics are usually not written with capital letters, and gentamicin is spelled with an "i" and not a "y".
Table 1.: It is surprising that patient no. 11 with a nephrostomy had a lower UTI ? In the Recurrent UTI line, what does "NA" mean?
P.5, line 6: 38 deg. Celcius, capital C
Page 2, first line: Sounds strange with "UTIs are one of", better either UTI not in pleural or: UTIs are among the most...
P. 2 line 7 from above: "antimicrobial bacterial resistande" - delete "bacterial"
P. 2, section 1.2, line 6: References 11,12 and 13 without all parentheses: (11,12,13).
P.2, section 1.2, line 12: How can patient´s family members provide "better" care than hospital care?
Comments on the Quality of English Language
As exemplified above, can be improved
Author Response
On behalf of all authors I thank you for your professional review that siginificantly improve this important manuscript.
Prof. Gad Segal, MD

Reviewer 2 Report
Comments and Suggestions for Authors
Author Response
On behalf of all authors, I thank you very much for your professional review and support.
Prof. Gad Segal, MD

Round 2
Reviewer 1 Report
Comments and Suggestions for Authors
The authors have adequately responded to the comments of the editor. The only comment from here is the introduction, where UTI now in singular should be folloaed by "is" not "are".
Comments on the Quality of English LanguageOk